# Structure and Activities of the NS1 Influenza Protein and Progress in the Development of Small-Molecule Drugs

**DOI:** 10.3390/ijms22084242

**Published:** 2021-04-19

**Authors:** Hyeon Jin Kim, Mi Suk Jeong, Se Bok Jang

**Affiliations:** 1Department of Molecular Biology, College of Natural Sciences, Pusan National University, Jangjeon-dong, Geumjeong-gu, Busan 46241, Korea; khjkhj0903@naver.com; 2Institute for Plastic Information and Energy Materials and Sustainable Utilization of Photovoltaic Energy Research Center, Pusan National University, Jangjeon-dong, Geumjeong-gu, Busan 46241, Korea

**Keywords:** NS1, influenza virus A, small molecule, drug, inhibitor

## Abstract

The influenza virus causes human disease on a global scale and significant morbidity and mortality. The existing vaccination regime remains vulnerable to antigenic drift, and more seriously, a small number of viral mutations could lead to drug resistance. Therefore, the development of a new additional therapeutic small molecule-based anti-influenza virus is urgently required. The NS1 influenza gene plays a pivotal role in the suppression of host antiviral responses, especially by inhibiting interferon (IFN) production and the activities of antiviral proteins, such as dsRNA-dependent serine/threonine-protein kinase R (PKR) and 2′-5′-oligoadenylate synthetase (OAS)/RNase L. NS1 also modulates important aspects of viral RNA replication, viral protein synthesis, and virus replication cycle. Taken together, small molecules that target NS1 are believed to offer a means of developing new anti-influenza drugs.

## 1. Introduction

Influenza viruses cause respiratory infections that seriously affect public health, typically they manifest as seasonal epidemics and intermittent pandemics that result in thousands of deaths [1]. The World Health Organization has estimated that annually, influenza causes 600,000 hospitalizations and 40,000 deaths in the United States. The worst influenza pandemic, the Spanish influenza pandemic (1918^H1N1^), occurred in 1918 and in eight months caused 50–100 million deaths worldwide [2]. Although avian H5N1 has not as-yet achieved person-to-person transmissibility, the death rate among the small number of people infected by direct contact with birds was as high as 60% [3]. On the other hand, swine flu (H1N1), which caused a pandemic in 2009, has high transmissibility and low virulence [4]. For these reasons, control of seasonal influenza presents a weighty challenge.

The viruses are enveloped viruses of the family *Orthomyxoviridae* and the surface glycoproteins haemagglutinin (HA) and neuraminidase (NA) define the viral subtypes and the immunogenic proteins of the influenza virus. They possess a segmented, negative-strand RNA genome consisting of eight segments of viral RNA encoding 11 proteins. These viruses are classified as influenza A, B, and C types. The genomes of A and B type influenza consist of eight single-stranded negative-sense RNA segments, whereas that of influenza C viruses has only seven [5,6]. Type A influenza viruses are responsible for most symptomatic infections in man, but they infect multiple species, including swine, dogs, cats, horses, humans, and other mammals. In contrast, type B influenza viruses exclusively infect humans. Influenza C viruses showed seasonal activity in pigs and are transmitted between pigs [7,8,9].

To survive in nature, influenza viruses have evolved multiple mechanisms to evade host immune systems. A small number of drugs have been approved for the treatment of influenza. Admantane antivirals target the viral M2 ion channel required for viral uncoating within host cells and inhibit viral neuraminidase protein. Recently, the FDA approved baloxavir marboxil, which targets viral polymerase [10,11]. However, clinical concerns remain regarding its efficacy, resistance, and cost [12]. Thus, it appears that effective responses to pandemics will require an extensive range of antiviral drugs.

## 2. Structure of the NS1 Protein

Nonstructural protein 1 (NS1) is a novel attractive antiviral target. Genetic analyses of NS1 have shown that it is involved in several processes of viral replication, pathogenesis, spread, and evasion of the host cellular innate immune system [13,14,15,16,17,18,19,20]. NS1 is encoded by all strains of influenza A and is well conserved [21]. NS1 protein lengths vary between strains from 230 to 237 aa [5]. NS1 accumulates in cell nuclei during early infection stages and later enters cytoplasm. This protein has two functional domains, that is, an *N*-terminal RNA-binding domain (RBD, residues 1–73) and a *C*-terminal effector domain (ED, residues 88–202), which are joined by a short interdomain linker region (LR). The remaining residues of NS1 that form the *C*-terminal tail (CTT) are intrinsically disordered, and the conformation of this region can change when NS1 interacts with host cell proteins or due to post-translational modifications [22]. RBD dimer consists of three α-helices from one monomer that is interlocked with three α-helices from the other in a six-helical symmetrical arrangement [23]. K41 and R38 of NS1 are essential for double-strand RNA (dsRNA) binding. Furthermore, whereas the K41A mutation reduces NS1 affinity for dsRNA, the R38A mutation entirely abrogates dsRNA binding [24]. The flexible LR of NS1 has a short type I β-turn encompassing residues 74 to 77 and acts as a mechanical hinge that allows the RBD and ED regions to move relative to one another [25]. Each ED monomer has seven β-strands and three α-helices. CTT has been used to determine NS1 structure, but the full-length structure has not been resolved (Figure 1).

The RBD domain binds dsRNA with low-affinity and protects the virus against the antiviral state induced by interferon α/β [26,27,28]. Binding of the RBD of NS1 with retinoic acid-inducible gene-I (RIG-I; a cytoplasmic pathogen sensor) inhibits the viral detecting ability of host cells [29]. The *C*-terminus of NS1 predominantly interacts with host cell proteins, such as elongation initiation factor 4G1 (eIF4G1), dsRNA-dependent serine/threonine-protein kinase R (PKR), and poly(A)-binding protein II (PAB II) and cleaves the 30 kDa subunit of polyadenylation specificity factor (CPSF30) [30,31,32,33]. NS1 is a constitutive homodimer, and its dimerization is mediated by high-affinity interactions between RBDs, whereas multimerization depends on ED via W187 [34,35]. Dimerization is essential for 1:1 binding between dsRNA and NS1-dsRNA complex [26,36]. NS1 is mainly localized to nuclei, though appreciable amounts can be found in cytoplasm [37]. NS1 contains a highly conserved nuclear localization sequence (NLS) in its RBD that utilizes Arg35, Arg38, and Lys41 to bind to dsRNA, although some virus strains possess a second NLS in the *C*-terminal. Concurrent with the *C*-terminus of NLS is a functional nucleolar localization signal (NoLS) [38], and a nuclear export signal (NES) lies within residues 138–147 [39].

## 3. Function of NS1 in Infected Cells

### 3.1. Translation of Viral mRNA

NS1 can bind with the 5′ untranslated region (5′UTR) of viral mRNAs, and this binding enhances viral translation, but not the translations of cellular mRNAs. Marion et al. reported that the *N*-terminal of NS1 plays a critical role in the stimulation of viral mRNA translation in COS-1 cells [40]. NS1 coimmunoprecipitates with eIF4G1 (a large subunit of eIF4F, residues 81–113) in transfected COS-1 cells (Figure 2) [41]. Additionally, NS1 residues 1 and 81 interact with PABPI residues 365 and 535 in an RNA-independent manner. Furthermore, these interactions suggest a NS1-PABPI-eIF4G1 heterotrimeric complex might be possible [32]. Falcon et al. reported hStaufen, which can bind with dsRNA and tubulin in interaction with NS1 in a two-hybrid genetic trap [42]. NS1 recruits eIF4G1, PABP, and hStaufen to the 5′ UTR of viral mRNA, which promotes viral mRNA translation. Furthermore, SUMOylation (small ubiquitin-like modifier conjugation) of K219 and K221 of NS1 accelerates viral replication and slightly increased virus titers of the highly pathogenic H5N1 strain in WSN-transfected cells. In contrast, ISGylation (also termed ISG15 modification) of K41 disrupted the interaction between NS1 and importin α, which interrupted the nuclear import of NS1 and reduced viral replication [43,44].

### 3.2. Responses of the Human Immune System

Host cells have the potential to prompt response to virus infection by activating the innate interferon (IFN) system, which can limit virus replication and spread. NS1 has shown to be a general inhibitor of the interferon signaling pathway, and infection with NS1 deletion virus was reported to induce large amounts of IFN-stimulated reporter gene in 293 cells at virus titers 10–100 fold lower than wild-type virus [18]. Influenza PR8/NS1 prevented virus- and/or the dsRNA-mediated activations of IFN regulator factor 3 (IRF3) and NF-κB, which is essential for the interferon system [45,46]. Upon viral infection, RIG-I recognizes viral RNA in a 5′-triphosphate-dependent manner and initiates an antiviral signaling cascade for MAVS/VISA/IPS-1/Cardif. Ubiquitin ligase tripartite motif 25 (TRIM25) mediates the K63-linked ubiquitination of RIG-I, which is crucially required for the RIG-I signaling pathway, and PR8/NS1 residues E96 and E97 directly interact with the coiled-coil domain of TRIM25, and thus, interfere with TRIM25 oligomerization and RIG-I CARD domain ubiquitination [47]. NS1 also post-transcriptionally regulates the expression of genes via interaction with the 3′ ploy(A) tail of mRNA. Qiu and Krug reported that NS1 protein inhibits the nuclear export of mRNAs containing 3′ poly(A) and concluded that NS1 is directly involved in the inhibition of host mRNA maturation via the interaction between NS1 and the poly(A) tail OF mRNA [48]. CPSF30 is an essential component of the mammalian pre-mRNA 3′ end processing mechanism. Ud/NS1 protein directly binds to CPSF30 to prevent 3′ end cleavage and polyadenylation of host pre-mRNA. Interaction between CPSF30 and NS1 inhibits 3′ end formation and the nuclear export of poly(A)-containing mRNAs [33]. Also, PAB II is required for the cleavage of host mRNA and plays a role in nuclear export. PAB II function is blocked by NS1 binding. Chen showed that individual NS1 protein forms complexes with CPSF30 and PAB II in virus-infected cells. By blocking the nuclear export of cellular host mRNA, NS1 shuts off cellular gene expression and non-processing of the 3′ end. In contrast, the nuclear export of viral mRNA is not obstructed, because the poly(A) tails of viral mRNAs are not synthesized by the cellular processing apparatus [30]. Some NS1 proteins escape the antiviral effects of IFNs and TNF-α. For example, H5N1/97 NS1 showed resistance to IFNs, TNF-α, and cytokines in SJPL cells [49].

Upon stimulation, the innate immune system is activated in many ways. Maturate dendritic cells (DCs) release proinflammatory cytokines and chemokines and expand innate lymphocyte populations. Interleukin (IL)-12 and type I IFNs stimulate the adaptive immune system of lymphocytes, the polarization of T helper type 1 (Th1) CD4^+^ T cells, and the development of cytotoxic T cells [50]. PR8/NS1 reduces the expressions of specific genes involved in DC maturation and migration, and NS1 protein acts as a bifunctional viral immunosuppressor that inhibits the innate IFN system and the adaptive immune system by interfering with host DC maturation and the activation of T-cell responses [51].

H3N2 NS1-tail can bind with PAF1. Human PAF1 complex (hPAF1C) consists of PAF1, LEO1, CDC73, SKI8, CTR9, and RTF1. hPAF1C mediates transcriptional elongation of antiviral and pro-inflammatory genes, and loss of hPAF1C by NS1 attenuates antiviral gene expression and increases vulnerability to virus infections [52].

### 3.3. OAS and PKR

NS1 can directly inhibit the functions of cytoplasmic antiviral proteins such as 2′-5′-oligoadenylate synthetase (OAS) and PKR. These two proteins were studied in a host IFN-induced antiviral and antiproliferative response system. dsRNA activated OAS synthesized 2′-5′ oligoadenylates, and thus, stimulated latent RNase L, leading to viral RNA degradation. IFN-β induced antiviral activity after RNase L activation activated OAS dependent on dsRNA. NS1 binding to dsRNA can inhibit the interaction between dsRNA and OAS [53,54]. NS1 also binds to PKR, a crucial host antiviral response protein. After PKR activation following binding with dsRNA or PACT protein, PKR undergoes a conformational rearrangement and autophosphorylation, which leads to the phosphorylation of the α subunit of eukaryotic initiation factor-2 (eIF-2α). Phosphorylation of eIF-2α impedes the initiation of viral and cellular protein synthesis. PKR also contributes to IFN response by activating NF-κB and IRF1. NS1 binds to the linker region of PKR and prevents the conformational change required for PKR autophosphorylation [31,55,56,57].

### 3.4. The Host RNAi Pathway and Apoptotic Response

RNA interference (RNAi) is well-conserved in eukaryotes. RNAi acts as a defense response against invading nucleic acids such as viruses. NS1 has been shown to inhibit RNAi in plants and *Drosophila melanogaster* [58,59]. However, NS1 from the A/WSN/33 strain did not suppress RNAi in mammalian cells [60]. These results suggest that NS1 from different strains might act via mechanisms not involving the RNAi pathway.

The role of NS1 in apoptosis has not been clarified, as NS1 has both pro- and anti-apoptotic functions. Limiting apoptosis early during virus infection might promote viral genome replication, while limiting it later can promote the propagation of influenza viruses and viral proteins. Autophosphorylation of PKR plays a key role in apoptosis during virus infection, and direct binding between PKR and NS1 can block PKR-mediated cell death [37,57]. As described below, NS1 also limits apoptosis by activating the host cell phosphatidylinositol 3-kinase (PI3K) pathway. The NS1 of influenza/swine/Colorado/1/1997 might induce apoptosis. NS1 interaction with heat shock protein 90 (Hsp90) induced the vulnerable interaction between Hsp90 and Apaf-1 but promoted the interaction between Apaf-1 and cytochrome c (Cyt c). These interactions activated caspase-9- and 3-related apoptosis in transfected A549 cells [61]

### 3.5. PI3K Signaling Pathway

PI3K consists of the regulatory subunit (p85) and 110 kDa enzymatic (p110) regulation downstream effector Akt/protein kinase B (PKB). The PI3K pathway controls essential processes in cells, such as metabolic regulation, growth, anti-apoptosis, proliferation, and cytokine production. Despite the PI3K-induced activation of IRF3, PI3K upregulates virus titers during the late stage of the virus replication cycle. Lack of NS1 protein failed to activate the PI3K/Akt pathway and inhibition of PI3K signaling resulted in impaired viral propagation in MDCK and A549 cells [62,63]. Y89 of NS1 is highly conserved and interacts with the p85β regulatory isoform of PI3K, and the interaction between NS1 and p85β activates PI3K. Ud/NS1 containing the Y89F mutation grew more slowly in tissue culture than the wild-type virus, which suggested that PI3K signaling plays an important role in virus replication in infected cells [64]. Heikkinen et al. reported that the NS1 proteins of birds, but not humans, enhance PI3K activation via the interaction between NS1 and Crk/CrkL. Details of cellular signaling involving NS1/PI3K/Crk remain to be clarified but this pathway could provide a useful target for novel anti-influenza drugs [65].

## 4. Virulence of NS1 in Influenza A Viruses

Influenza NS1 protein is a multifunctional protein that counteracts host immune response, and thus, allows the virus to replicate efficiently in IFN response. Studies have demonstrated that the NS1 gene is important for the virulence of several subtypes of influenza in various species. For instance, the virus acquired the virulent of the NS1 gene GS/GD/1/96 virus, which contains Ala 149. This mutation resulted in the antagonization of interferon induction in chicken embryo fibroblasts and proved highly pathogenic in chickens [66]. Deletion of residues 191 to 195 of NS1 in the SW/FJ/03 virus resulted in low pathogenicity and an inability to antagonize host IFN response in chickens. Deletion of these residues affected the ability of NS1 to bind to CPSF30 [67]. WSN viruses containing the S42G mutation exhibited increased viral replication and inhibited INF production during viral infection, but this was not accompanied by dsRNA binding ability [16]. *C*-terminally truncated NS1 virus in attenuated H7N7 virulence in mammalian and avian cells [68], and carboxy-truncated NS1 in TX/98 viruses, were less able to suppress the INF system in pig cells [69]. Seo et al. reported that H5N1/97 virus, which contains glutamic acid at amino acid position 92 of NS1, exhibited resistance to the antiviral effects of INF and TNF-α [49]. In addition, deletion of NS1 263 to 277, which are associated with D92E sift, in H5N1 viruses significantly increased virulence and replication in chickens and mice, and deletion of 5 amino acids, from 80 to 84, resulted in D92E sift in H5N1 and increased virulence [70]. A recent large-scale genome sequence analysis found that *C*-terminal residues of avian influenza virus NS1 proteins have a PDZ ligand domain of the X-S/T-X-V type. Proteins with a PDZ domain play important roles in the localization, transport, and assembly of signaling complexes. Binding of the NS1 *C*-terminus in the presence of PDZ-binding proteins resulted in increased virulence, pathogenesis, and IFN-antagonism, and highly pathogenic avian influenza (HPAI) virus-containing NS1 *C*-terminal residues (ESEV or EPEV) showed a remarkable increase in virulence and pathogenicity in infected mice but did not influence virus growth [19].

## 5. Antiviral Compounds Targeting NS1

### 5.1. Small Molecules Targeting NS1

dsRNA is essential for NS1 dimerization and activation. Thus, the dsRNA and NS1 interaction is a potential target for small-molecule inhibition. Structural and mutational analysis showed that the R38 of the RBD of NS1 is vital for dsRNA binding [16]. Epigallocatechin gallate (EGCG) significantly decreased viral NS1 level and viral replication, and a thermal denaturation assay showed that EGCG might also interact with NS1 at its dsRNA binding motif R38. Also, epicatechin gallate (ECG) effectively inhibited viral replication in infected MDCK cells [71]. Quinoxaline and EGCG have similar bicyclic ring structures. Compound 44 derived from quinoxaline inhibited Ud/72 virus growth ~10-fold in infected MDCK cells. Compound 44 did not interfere with the interaction between NS1 and dsRNA, but likely functioned by binding to the NS1 dsRNA-binding motif [72]. JJ3297 and A9 interact with the ED of NS1 in a hydrophobic pocket known to bind CPSF30. W187 in ED is required for dimer formation and the forms have a hydrophobic pocket for binding to CPSF30. Interfering with this interaction might provide a useful strategy [73]. JJ3297 reduced virus replication ten-fold and produced significant levels of IFN-β in infected cells in the presence of RNase L, which is a critical downstream effector of IFN. Moreover, JJ3297 facilitated the induction of an IFN-like antiviral state in infected cells. However, RNase L^−/−^ MEF cells were entirely resistant to JJ3297 treatment in mouse embryo fibroblast (MEF) cells infected with A/PR/8 viruses [74]. Drug development based on JJ3297 resulted in compound A22. Chemical shift perturbation (CSP) showed that A22 interacts with 1918^H1N1^ and Shaghai/1/2013^H7N9^ NS1 ED. Furthermore, 2013^H7N9^ has spread in China and resulted in 1500 human infections with a 40% mortality rate [75]. Kleinpeter et al. reported that A22 interacts with the CPSF30-binding pocket and interrupted NS1-CPSF30 binding in a diverse strain of IAV and suggested it might be an effective antiviral [73]. In addition, molecular mechanics Poisson-Boltzmann surface area (MM-PBSA) calculations suggested that the potential anti-influenza inhibitors, 30256 and 31674, stably bind to the CPSF30-binding site in NS1 with high binding-free energy. These two compounds can also form strong hydrophobic interactions with the minor pocket of NS1 near W187. [76]. Basu et al. presented four compounds that suppress NS1 function and effectively inhibited viral replications in Hong Kong/19/68-, WSN/33-, and PR/8 IAV-infected MDCK cells. In addition, three of these compounds (NSC128164, NSC109834, and NSC95676) strongly reduced viral M2 protein and RNA levels. However, despite NSC125044 significantly increasing IFN-β mRNA levels, it did not affect viral RNA levels [77]. NS1 antagonists derived from pyrazolopyridine inhibited PR/84 virus replication in MDCK cells. Compound 32 was a potent antiviral, as determined by IFN-β mRNA levels, and showed no sign of treatment accumulation or toxicity in any organ [78]. Mata et al. identified an NS1 inhibitor by high-throughput screening. Compound 3 derived from naphthalimides was much less cytotoxic and had a considerably longer half-life than the other compounds examined. In addition, compound 3 reduced viral protein levels and replication but did not induce IFN production or mediated response. Notably, compound 3 did not show an antiviral effect in the absence of REDD1. Influenza viruses stimulate mTOR complex 1 (mTORC1) signaling through the activation of Akt via T308 phosphorylation. The mTORC1 inhibitor REDD1 induced the dephosphorylation of Akt T308 in infected cells. REDD1^−/−^ MEF cells produced large amounts of influenza virus proteins 2–3 h earlier and were highly permissive to virus replication. These results suggested that the induction of REDD1 by naphtalimides offers a novel means of inhibiting the mTORC1 pathway [79,80]. Baicalin-derived flavonoids residues Q40 and S42 interacted with NS1 in its 39–43 amino acid region, disrupted the NS1–p85β interaction, and thus, reduced Akt phosphorylation. Baicalin reduced virus replication and viral NP transcription in A549 cells infected with H1N1/PR8. Also, the expressions of IFN-α and IFN-β were time-dependently increased in the presence of baicalin. In contrast, the expressions of the antiviral genes RIG-I and PKR were noticeably decreased in cells treated with baicalin [81]. H5N1 avian influenza virus has caused infections in birds and poultry globally, and in chickens, has a mortality rate of up to 87.5%. The widespread nature of viral infections presents a potential pandemic threat to humans. Antisense oligonucleotides might be useful drugs for treating influenza viral infections as base-pairing between viral DNA or RNA and oligonucleotides might interfere with viral transcription and replication. Wu et al. found that treatment with an oligonucleotide targeting NS1 significantly inhibited viral replication in chicken embryo fibroblast cells and protected chickens from virus-induced death [82]. Influenza virus inhibitors are summarized in Table 1.

### 5.2. Other Small Molecules Targeting Influenza Virus A

Receptor tyrosine kinase (RTK) signaling triggers diverse pathways, such as the Ras/ERK/MAPK, JAK/STAT3, and PI3K/Akt pathways, which are associated with cell growth, metabolism, differentiation, and migration. RTKs also play important roles in virus replication. RTK inhibitors (RTKIs), such as AG879 and tyrphostin A9, have strong antiviral actions against influenza A. Viral ribonucleoprotein (vRNP) complex can be exported to cytoplasm via the cellular Crm-1-mediated nuclear export pathway. In A549 cells infected with A/WSN viruses, treatment with AG879 or A9 directly inhibited the Crm1 nuclear export pathway, which may have resulted in the accumulation and nuclear retention of influenza vRNPs. In addition, AG879 and tyrphostin A9 inhibit viral RNA synthesis and the release of influenza virus particles independently of the NF-κB pathway. Thus, TrkA signaling is crucially required for influenza virus replication. Three TrkA inhibitors (AG879, GW441756, and K252a) suppressed influenza A virus replication in 549 cells infected with A/WSN viruses [83]. Cellular nucleotides containing pyrimidines and purines play vital roles in RNA and DNA. Quinoline acid targets dihydroorotate dehydrogenase (DHODH), a host enzyme required for pyrimidine biosynthesis, which interferes with pyrimidine production and limits virus replication. Furthermore, a DHODH inhibitor suppressed virus protein levels and virus replication and reduced mRNA nuclear export mediated by NS1 [84].

## 6. Conclusions

Influenza is an acute global public health problem that causes considerable morbidity and mortality. About 5–15% of the population are infected by influenza A viruses. Influenza viruses are highly contagious and pathogenic and cause serious respiratory disease in humans and diverse animal species. Currently, research efforts directed against the virus focus on preventative vaccines and antiviral drugs. Vaccination has been a highly effective strategy in terms of reducing the number of infections and deaths caused by influenza viruses. However, viral antigenic drift dictates that novel vaccines be developed annually, and the effectiveness of these vaccines vary. In addition, antiviral agents against influenza have been developed that target the viral M2 ion channel and inhibit viral neuraminidase and polymerase, but the continuous increase in the number of resistant strains means that these drugs may be ineffective after a few years due to the appearance of new strains. Although the resistance of drugs remains unrevealed, the increasing emergence of various drug-resistant influenza strains play up the need for continuous development of improved antiviral drugs and strategies, including rational use of currently available antivirals, drug combinations, and supervising influenza virus drug resistance. This situation highlights the need for additional anti-influenza drugs that specifically target viral processes. NS1 is encoded by all strains of influenza and is a highly conserved protein that plays a vital role in evading host antiviral response. NS1 is a dsRNA-binding protein that can interact in diverse ways with cellular proteins. Binding of NS1 and dsRNA inhibits the OAS/RNase pathway of viral RNA degradation, and the PKR-dependent phosphorylation of eIF2-α is inhibited by direct binding between NS1 and PKR. NS1 inhibits 3′-end processing of cellular pr-mRNAs and nuclear exporting by binding to CPSF30 and PAB II. Additionally, NS1 acts to inhibit IFN-mediated host antiviral response. NS1 blocks the induction of IFN gene transcription and mRNA maturation, and it was recently demonstrated that NS1 interacts with TRIM25, prevents RIG-I ubiquitination, and thus, inhibits the synthesis of cellular IFN. Other functions of NS1 include the induction of viral gene expression and the mediation of host cell apoptosis. Therefore, NS1 presents a high-value target for the development of antiviral compounds that control viral replication and spread.

## Figures and Tables

**Figure 1 ijms-22-04242-f001:**
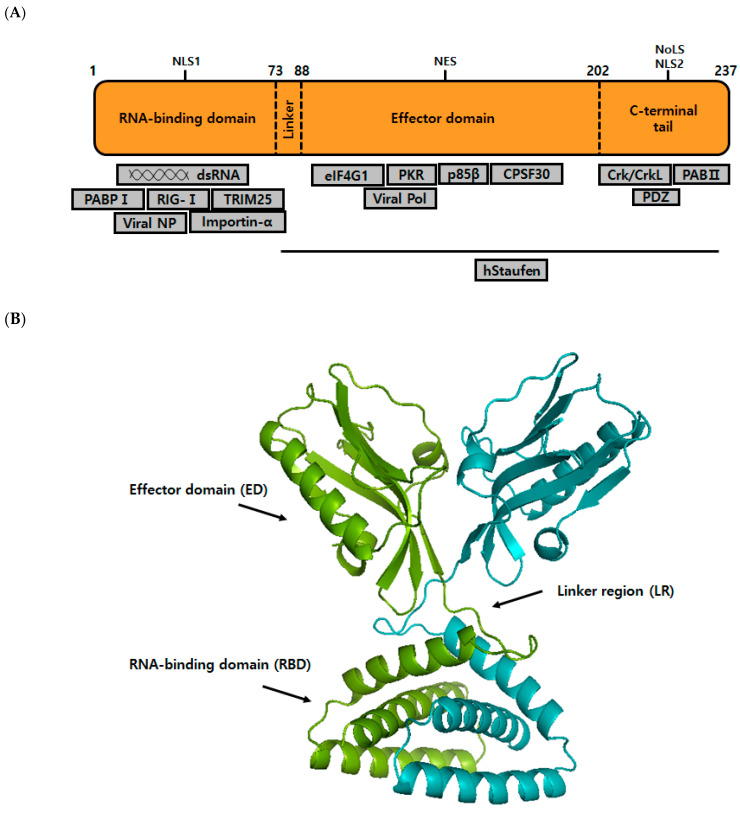
Structural analyses of NS1. (**A**) Schematic representation of the full-length NS1 domain and of interacting proteins. NS1 consists of an RNA-binding domain (RBD), a linker region (LR), an effector domain (ED), and a *C*-terminal tail (CTT). RBD binds to dsRNA to inhibit the interferon (INF) induction, whereas the *C*-terminal ED region mediates interactions with several host cellular proteins. (**B**) NS1 dimer adapted from the X-ray structure of H6N6 NS1 (PDB ID: 4OPA), showing interactions between the two RBD.

**Figure 2 ijms-22-04242-f002:**
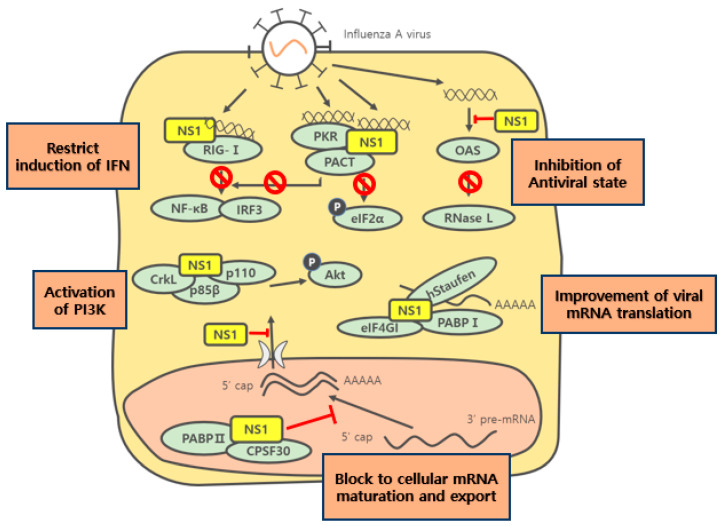
Overview of the functions of NS1 protein in cytoplasm and nuclei of infected cells. NS1 plays a vital role as an INF antagonist for the influenza virus. NS1 inhibits IFN induction at the pre-transcriptional stage by blocking the activation of the RIG-I and at the post-translational level by repressing activations of the antiviral properties of PKR and OAS/RNase L. Furthermore, NS1 disrupts the processing and nuclear export of cellular mRNAs. NS1 enhances viral mRNA translation, and activation of the PI3K pathway by NS1 is required for viral replication in infected cells.

**Table 1 ijms-22-04242-t001:** Small-molecule inhibitors of influenza A.

Molecules	Targeting	Model Studied	Effects	Ref.
**EGCG**	NS1	MDCK cells	Decreased the viral NS1 level and viral replication	[71]
**Quinoxaline**	NS1	MDCK cells	Inhibiting viral replication	[72]
**JJ3297**	NS1	MEF cells	Reduced virus replication and produce level of IFN	[74]
**A22**	NS1	NMR	Interrupt the NS1-CPSF30 binding	[75]
**Compound 32056, 31674**	NS1	MM-PBSA	Binding with NS1 near the W187	[76]
**NSC 109834, 128164, 95676, 125044**	NS1	MDCK cells	Reduced viral M2 protein and RNAs	[77]
**pyrazolopyridine**	NS1	MDCK cells	Inhibition of virus replication	[78]
**Compound 3**	NS1	MDCK cells	Decreased viral protein and replication	[79,80]
**Baicalin**	NS1	A549 cells	Reduced virus replication and viral NP transcription	[81]
**RNA oligonucleotide**	NS1	chicken embryo fibroblast cells	Interference to the viral transcription and replication	[82]
** AG879, tyrphostin A9 **	Receptor tyrosine kinases	A549 cells	Inhibit the Crm1 nuclear export pathway and viral RNA synthesis	[83]
** Quinoline acid **	dihydroorotate dehydrogenase	MDCK cells	Inhibition of production pyrimidines that limits virus replication	[84]

EGCG: Epigallocatechin gallate; NS1: Non-structural protein 1; MDCK: Madin-Darby Canine Kidney; IFN: Interferon; MEF: Mouse embryonic fibroblast; MM-PBSA: Molecular mechanics Poisson-Boltzmann surface area.

## Data Availability

Not applicable.

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
