# Peer review of "Structure and Activities of the NS1 Influenza Protein and Progress in the Development of Small-Molecule Drugs"

_ijms, 2021, doi:10.3390/ijms22084242_

Round 1

Reviewer 1 Report

In the current manuscript, Kim et al provide a detailed review of the structure and functions of the influenza NS1 protein, and provide insight into the development of small-molecule drugs targeting NS1. The information provided logically builds upon the perspective the authors provided in an earlier review on the topic in 2019 (PMID: 31154753). The current review provides a thorough description of the structural domains of NS1, which is key to understanding of the activity of the small-molecule drugs mentioned later in the manuscript. Additionally, the authors provide an organized and clearly written summary of the many immune evasion functions attributed to NS1. My recommended corrections and additions are:

  1. Page 2, section 2, last paragraph, line 4: recommend use of "C-terminus" rather than "C-terminal".
  2. Page 4, line 1: the sentence as written is confusing and the comma appears misplaced; are the authors intending to state that hStaufen has the ability to bind dsRNA and tubulin, and that a two-hybrid genetic trap revealed that hStaufen can also interact with NS1?
  3. Page 4, section 3.2, line 1: correct to interferon (IFN)
  4. Page 5, line 17: correct to "...antiviral effect of IFNs..."
  5. Page 5, paragraph 2, line 7: correct to "innate IFN system..."
  6. Table 1: it may be useful to readers to include a column after "Effects" to list the model system used in the referenced study (cell line, egg, lab animal) to provide some idea of how far along in testing are these molecules

Author Response

In the current manuscript, Kim et al provide a detailed review of the structure and functions of the influenza NS1 protein, and provide insight into the development of small-molecule drugs targeting NS1. The information provided logically builds upon the perspective the authors provided in an earlier review on the topic in 2019 (PMID: 31154753). The current review provides a thorough description of the structural domains of NS1, which is key to understanding of the activity of the small-molecule drugs mentioned later in the manuscript. Additionally, the authors provide an organized and clearly written summary of the many immune evasion functions attributed to NS1. My recommended corrections and additions are:

  1. Page 2, section 2, last paragraph, line 4: recommend use of "C-terminus" rather than "C-terminal".

It has been modified.

  1. Page 4, line 1: the sentence as written is confusing and the comma appears misplaced; are the authors intending to state that hStaufen has the ability to bind dsRNA and tubulin, and that a two-hybrid genetic trap revealed that hStaufen can also interact with NS1?

It has been modified.

  1. Page 4, section 3.2, line 1: correct to interferon (IFN)

It has been modified.

  1. Page 5, line 17: correct to "...antiviral effect of IFNs..."

It has been modified.

  1. Page 5, paragraph 2, line 7: correct to "innate IFN system..."

It has been modified.

  1. Table 1: it may be useful to readers to include a column after "Effects" to list the model system used in the referenced study (cell line, egg, lab animal) to provide some idea of how far along in testing are these molecules

A table has been added to the text.

Reviewer 2 Report

The review "Structure and activities of the NS1 influenza protein and progress in the development of small-molecule drugs" highlights the importance of NS1 protein in Influenza virus replication cycle and points to the importance of targeting this protein with small molecules to inhibit virus replication.

The review has important information and details regarding this topic. It is well-written and will be an add to the field.

The main point that I would suggest is to also talk about the possibility of development of resistant influenza strains after the use of small molecules targeting NS1.

Author Response

The main point that I would suggest is to also talk about the possibility of development of resistant influenza strains after the use of small molecules targeting NS1.

It has been added to the conclusion of the text.